# DEALING OUT OF DISTRIBUTION WITH PREDICTION PROBLEM

## ABSTRACT

The open world assumption in model development means that a model may lack sufficient information to effectively handle data that is completely different or out of distribution (OOD). When a model encounters OOD data, its performance can significantly decrease. Improving the model's performance in dealing with OOD can be achieved through generalization by adding noise, which can be easily done with deep learning. However, many advanced machine learning models are resource-intensive and designed to work best with specialized hardware (GPU), which may not always be available for common users with hardware limitations. To provide a deep understanding and solution on OOD for general user, this study explores detection, evaluation, and prediction tasks within the context of OOD on tabular datasets using common consumer hardware (CPU). It demonstrates how users can identify OOD data from available datasets and provide guidance on evaluating the OOD selection through simple experiments and visualizations. Furthermore, the study introduces Tabular Contrast Learning (TCL), a technique specifically designed for tabular prediction tasks. While achieving better results compared to heavier models, TCL is more efficient even when trained without specialised hardware, making it useful for general machine-learning users with computational limitations. This study includes a comprehensive comparison with existing approaches within their best hardware setting (GPU) compared with TCL on common hardware (CPU), focusing on both accuracy and efficiency. The results show that TCL exceeds other models, including gradient boosting decision trees, contrastive learning, and other deep learning models, on the classification task.

## 1 INTRODUCTION

The concept of open-world assumption in model development means that a model may not have enough information to effectively handle data that is completely different or out of distribution (OOD). When a model meets OOD data, it may suffer a significant decrease in performance (Hsu et al., 2020; Hendrycks and Gimpel, 2016). To handle this, model generalisation by introducing noise can be used, which can be achieved easily with deep learning. However, advanced deep learning algorithms such as FT-Transformer benefit from the advancement of specialised hardware such as GPU or TPU (Hwang, 2018), this type of hardware is not always available to the general user (Ahmed and Wahed, 2020). These demand the user to find the best way to deal with these challenges and emphasize the importance of our research.

While out-of-distribution (OOD) detection has been extensively studied (Lee et al., 2020a), the challenge of prediction tasks for OOD data, particularly in tabular datasets, remains underexplored. Significant progress has been made in OOD detection with algorithms like MCCD (Lee et al., 2020b), OpenMax (Bendale and Boult, 2016), Monte Carlo Dropout (Gal and Ghahramani, 2016), and ODIN (Liang et al., 2017). However, the study of prediction tasks on OOD for tabular data is limited. Tree-based classical models are known to

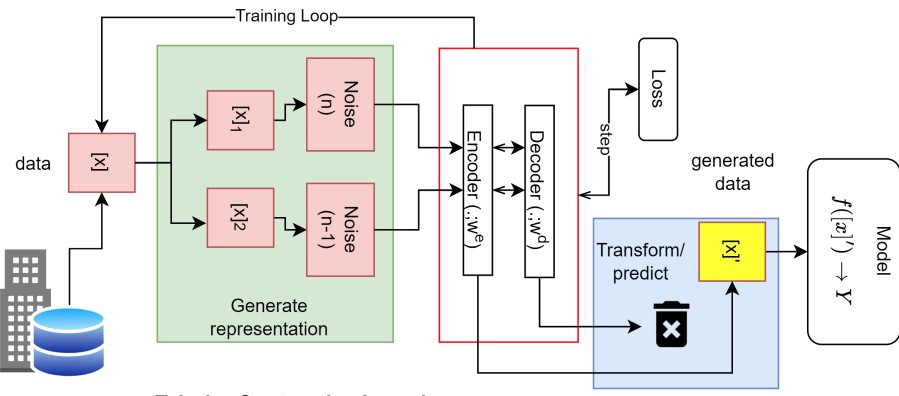

Figure 1: Tabular Contrastive Learning (TCL). The data $[x]$ is duplicated $([x]_1, [x]_2)$ and noise is added. Both duplicates are then encoded and decoded to compute the loss. During inference, TCL only uses the encoder to produce new data $[x]'$ that enhances supervised learning performance $f([x]') \rightarrow Y$. The decoder is omitted during inference and used only for training.

be reliable for tabular data (Grinsztajn et al., 2022) , but our experiments show that these models exhibit a decrease in performance when dealing with OOD data.

In this study, we made several contributions. First, we show step by step how to implement existing methods for detecting, separating, evaluating, and visualizing out-of-distribution (OOD) data using real-world datasets. Second, we assess the performance of existing tabular machine learning algorithms in handling OOD data. Lastly, we introduce a new approach called TCL, which provides efficiency and flexibility while achieving comparable performance.

Tabular Contrast Learning (TCL), Figure 1, is a local adaptation of Contrastive Federated Learning (CFL) (Ginanjar et al., 2024) designed for prediction tasks on tabular datasets for a general user. TCL is based on the principles of contrastive learning (Chen et al., 2020; Ucar et al., 2021) but is optimized for tabular data structures. TCL approach offers several advantages, e.g. **Efficiency**: TCL is designed to be faster and more compact compared to current state-of-the-art models, **Flexibility**: TCL can be integrated with various supervised learning algorithms and **Performance**: TCL achieves competitive performance.

Our experiment demonstrates that TCL delivers performance and efficiency (Huang et al., 2017) (defined by a higher speed/accuracy trade-off score) compared to other models.

## 2 RELATED WORK

### 2.1 OOD DETECTION

**OpenMax** (Bendale and Boult, 2016) uses the concept of Meta-Recognition to estimate the probability that an input belongs to an unknown class. OpenMax characterizes the failure of the recognition system and handles unknown/unseen classes during operation. In deep learning, SoftMax calculates as $P(y = j|x) = \dfrac{e^{v_j(x)}}{\sum_{i=1}^{N} e^{v_i(x)}}$ OpenMax recognizes that in out of distribution (OOD), the denominator of the SoftMax layer does not require the probabilities to sum to 1.

**TemperatureScaling** (Platt et al., 1999) is a single-parameter variant of the Platt scaling. In a study by Guo *et al.* (Guo et al., 2017), despite its simplicity, temperature scaling is effective in calibrating a model for deep learning. This also suggests that temperature scaling can be used to detect OOD. Our study uses these two approaches to separate OOD from the dataset and use it as validation data.

**Multi-class classification, deep neural networks, Gaussian discriminant analysis (MCCD)** (Lee et al., 2020b) is OOD detection algorithm based deep neural network that claim to have better classification inference performance. It is focuses on finding sperical-decission across classes.

Our work mainly uses OpenMax and TemperatureScaling. While the original algorithms are not new, both algorithm have latest update and better support under pytorch-ood (Kirchheim et al., 2022) compared to MCCD (Lee et al., 2020c).

## 2.2 TABULAR DATA PREDICTION

**Neural Network-based Methods:** Multilayer Perceptron (MLP) (Ruck et al., 1990; Gorishniy et al., 2023): A straightforward deep learning approach for tabular data. Self-Normalizing Neural Networks (SNN). (Klambauer et al., 2017): Uses SELU activation to train deeper networks more effectively.

**Advanced Architectures:** Feature Tokenizer Transformer / FT-Transformer (Vaswani et al., 2017): Adapts the transformer architecture for tabular data, consistently achieving high performance. Residual Network / ResNet (Li et al., 2018): Utilizes parallel hidden layers to capture complex feature interactions. Deep & Cross Network / DCN V2 (Wang et al., 2020): Incorporates a feature-crossing module with linear layers and multiplications. Automatic Feature Interaction / AutoInt (Song et al., 2018): Employs attention mechanisms on feature embeddings. Neural Oblivious Decision Ensembles / NODE (Popov et al., 2019): A differentiable ensemble of oblivious decision trees. Tabular Network / TabNet (Arık and Pfister, 2019): Uses a recurrent architecture with periodic feature weight adjustments. Focuses on attention framework.

**Ensemble Methods:** GrowNet (Badirli et al., 2020): Applies gradient boosting to less robust MLPs, primarily for classification and regression tasks.

**Gradient Boosting Decision Tree (GBDT)** (Grinsztajn et al., 2022) : XGBoost :A tree-based ensemble method that uses second-order gradients and regularization to prevent overfitting while maximizing computational efficiency. LightGBM :A fast and memory-efficient boosting framework that uses histogram-based algorithms and leaf-wise tree growth strategy for faster training. CatBoost :A gradient boosting implementation specifically optimized for categorical features with built-in ordered boosting to reduce prediction shift.

These models have shown varying degrees of success in tabular data prediction. However, their performance on OOD data remains a critical area for investigation. We include mentioned model as our base models.

## 2.3 TABULAR CONTRASTIVE LEARNING

**SubTab** (Ucar et al., 2021) and **SCARF** (Bahri et al., 2022) are a contrastive learning model tailored for tabular datasets. Similar to the fundamental concept of contrastive learning for the image of SimCLR (Chen et al., 2020). SubTab and SCARF calculate contrastive loss using cosine or euclidean distance. **CFL** (Ginanjar et al., 2024) is a federated learning algorithm proposed to tackle vertical partition within data silos. CFL explores the possibility of implementing contrastive learning within vertically partitioned data without the need for data sharing. CFL merge the weight by understanding that the data came from global imaginary dataset which is vertically partitioned. CFL uses contrastive learning as a medium for black box learning. CFL focuses on collaborative learning across silos. In this study, we study learning from local data with OOD, a problem that is yet to be explored by CFL CFL focuses on a Federated learning network, while ours is common tabular data. CFL, while exhibiting a similar name, uses partial data augmentation as part of

the federated learning concept and is similar to image contrastive learning. TCL, in the other hand use full matrix augmentation to support tabular data.

## 3 PROBLEM FORMULATION

### 3.1 DEFINITION

**Definition 1: Tabular Data.** Let $D \in R^{nd}$ be a tabular dataset with $n$ samples and $d$ features, and $Y \in R^n$ be the corresponding labels. Tabular data is characterized by its structured format, where each row represents a sample and each column represents a feature.

**Definition 2: Out-of-Distribution Prediction.** Given a model trained on in-distribution data $D_{in} = \{(x_i, y_i)|x_i \in X_{in}, y_i \in Y_{in}\}$, where $X_{in}$ follows a distribution $p_{in}$, the task is to make accurate predictions on OOD data $X_{ood}$ that follows a different distribution $p_{ood}$, where $p_{ood} \neq p_{in}$.

**Definition 3: Efficiency-Accuracy Trade-off.** We use a speed/accuracy trade off (Huang et al., 2017) from the total performance matrix. Let $T$ be the total performance , then $T = \frac{P}{t}$ for classification or $T = \frac{1/P}{t}$ for regression.

A performance evaluation $P$ used is F1 or RMSE for the prediction task and the time $t$ in seconds for the duration of training. The $P$ is used to obtain the standard performance of a model. The $t$ is used to evaluate the time it takes to train a model. A smaller $t$ means a smaller resource to find the best model by tuning the hyperparameters. The adjustment $\frac{1}{P}$ for regression is necessary because in regression tasks, RMSE is used as the performance metric, and smaller values are better.

### 3.2 PROBLEM STATEMENT

In machine learning, the goal is to build a model $f : x \rightarrow y$ that generalizes unseen data. However, if the model is exposed to samples outside the distribution during inference, it may make unreliable predictions or exhibit unexpected behaviour.

Formally, a prediction task can be defined as finding a model $f : (.)$ that minimizes the expected error over a dataset $D_{in}$ with distribution $p_{in}$. This can be defined as **min** $\text{Error}(x, y)_D = [L(f(x), y)]$, where $L$ is a loss function that measures the discrepancy between the prediction of the model $f(x)$ and the true label $y$. A bigger $E$ means poor model performance $P$ or can be denoted as $E \uparrow = P \downarrow$. When a different distribution $p_{ood}$ is introduced to a model, the performance decreased or $P(f((x)_{D_{in}})) > P(f((x)_{D_{ood}}))$.

The time $t$ to find the best model should also be considered. A time-consuming model takes more resources to train and tune. When dealing with a large dataset, the time used to train and tune a model is a big concern. A larger and longer model does not always equal better performance $P$. This can be written as $t \uparrow \neq P \uparrow$.

We evaluated the total/overall performance of the models when dealing with tabular dataset with OOD. The overall objection can be writen as $T = \frac{\max P_{ood}}{\min t} = \frac{1/\min E(x,y)_{D_{ood}}}{\min t}$.

## 4 PROPOSED METHOD

We introduce Tabular Contrastive Learning (TCL), an improved approach designed to enhance prediction tasks on tabular data, particularly with hardware limitations.

## 4.1 TCL Main Architecture

**Augmented Data**

As a contrastive learning based algorithm, TCL works based on augmented data. TCL creates two data augmentations. Denotated as $\{x^1, x^2\} = \text{Aug}(x)$. Noise was added to these augmented data.

**Matrix Augmentation**

In TCL, all original data (without slicing or splitting) is utilized as a representation. Unlike previous approaches such as simCLR (Chen et al., 2020) and SubTab (Ucar et al., 2021) that use slice, TCL utilizes the entire original data matrix for representation. This allows for capturing more comprehensive feature interactions in tabular data.

**Encoder-Decoder Structure**

The contrastive learning architecture includes two main components: an encoder and a decoder. The encoder transforms input data into a compressed representation called the latent space, denoted $E : x; \omega^e \rightarrow x^e$ where $\omega$ is parameter and $e$ is encoder notation, while the decoder reconstructs the original data from this representation denoted $P : x^e; \omega^p \rightarrow x^p$ where $p$ is notation for decoder. During training, both components are used, but only the encoder is employed during inference, allowing efficient compression of new data into its learned latent representation without reconstruction.

**Modified Contrastive Loss**

Loss is calculated based on augmented data, not original data. TCL simplifies contrastive loss calculation to enhance both performance and training speed. In contrastive learning, the loss is computed based on the similarity or dissimilarity of the augmented noisy data. Since TCL deals with row-based tabular data and it is a unsupervised learning, the method used is similarity. TCL aims to pull the noisy data points that originated from the same data. This is achieved by minimizing the total loss $L_c$ between representations. During training, the total loss is calculated as follows:

$$L_t(x) = (L_r(x) + L_c(x) + L_d(x)) \tag{1}$$

Where $L_r$ is the reconstruction loss, $L_c$ is the contrastive loss, and $L_d$ is the distance loss. The objective of contrastive learning is to minimize the total loss $L_t$. When $D$ is dataset, and sliced data $\mathcal{B} \in D$ , then:

$$\min L_t(.; \omega^e, \omega^p) = \min \frac{1}{J} \Sigma_{j=1}^J L_t(P(E(.; \omega^e); \omega^p)) \tag{2}$$

with $w$ is weight, $P(.)$ is decoder function, and $E(.)$ is encoder function. When $MSE(.)$ is the mean square error function, and $[x]$ is noisy data of $\mathcal{B}$, then: $L_r(x) = \frac{1}{N} \sum_n^N MSE(\hat{x}, x)$ and $L_d(x) = \frac{1}{N} \sum_n^N MSE(x^{e1}, x^{e2})$. While $L_r$ is calculated from decoded data and original data, $L_d$ is calculated from encoded data only. We simplified the contrastive loss $L_c$ by using only the result of a dot product compared with other contrastive learning that used euclidean distance.

$$L_c(x) = \frac{1}{N} \sum_n^N (-log \frac{\exp(MSE([0], dot(x^{e1} \cdot x^{e2}))/\mathcal{T})}{\sum_{k=1}^K \exp(MSE([0], dot(x^{e1} \cdot x^{e2}))/\mathcal{T})}) \tag{3}$$

## 5 TCL Algorithm

The TCL process involves several steps. First, a minibatch of N samples is sampled from the dataset. Then, for each sample in the batch, two augmented views are created and added with noise. Although they came

from the same data, both augmented data are different due to previous treatments. These augmented views are passed through an encoder network to obtain encoded representations. A decoder is then applied to the encoded representations. The loss function then calculates the difference between two augmented data. By minimizing this loss, noisy data is pulled together. Because TCL applies full matrix representation, the process pulls noisy data row by row together. This results in generalized data for better inference. The complete steps are presented on Algorithm 2 in the Appendix section.

# 6 EXPERIMENT

## 6.1 DATASETS

We utilized 10 diverse tabular datasets. The datasets are Adult (Becker and Kohavi, 1996), Helena (Guyon et al., 2019), Jannis (Guyon et al., 2019), Higgs Small (Baldi et al., 2014), Aloi (Geusebroek et al., 2005), Epsilon (PASCAL Challenge on Large Scale Learning, 2008), Cover Type (Blackard and Dean, 2000), California Housing (Pace and Barry, 1997), Year (Bertin-Mahieux et al., 2011), Yahoo (Chapelle and Chang, 2011), and Microsoft (Qin and Liu, 2013).

## 6.2 OOD DETECTION

We have implemented two Out-of-Distribution (OOD) detection methods, namely OpenMax (Bendale and Boult, 2016) and TemperatureScaling (Platt et al., 1999). We applied OOD detection methods to each dataset to transform the data and establish thresholds. The thresholds were manually assigned by observing the graphs produced with the OOD detection algorithm. The manual assignment was done by selecting a single point on a tail of the observation. We then separated the OOD data based on the thresholds to generate two sets, $D_{in}^{M}$ and $D_{ood}^{N}$, where $M$ and $N$ are total sample in each set, $(D_{in} + D_{ood} = D)$. We expect to find $M > N$. The OOD separation is validated using linear regression. Finally, we compared the model performance with and without OOD separation, expecting to find that the performance decreased in this step $f : D_{in} > f : D_{ood}$.

The results of the experiment is two set of dataset $D_{in}, D_{ood}$. While $D_{in}$ is used for train dataset, $D_{ood}$ is used for test dataset. The Algorithm 1 in the Appendix section explains step by step the OOD detection is done.

## 6.3 PREDICTION ON OOD DATASET

We experimented with 12 based models. For deep learning tabular models we use FT-T, DCN2, GrowNet, ResNet, MLP, AutoInt and TabR-MLP (Gorishniy et al., 2023). In addition, we experimented with the recent implementation of contrastive learning for tabular data SubTab (Ucar et al., 2021) and SCARF (Bahri et al., 2022), which comes from a similar domain to our TCL. We also did not apply FT-Transformer to some datasets. The FT-Transformer is heavy and has reached our hardware limitation. Finally, we compared our TCL with GDBT models.

Our experiment used an NVIDIA H100 GPU for all models except TCL and the GBDT-based model. TCL and GBDT were trained on a CPU (Apple / AMD) to emphasize our advancement within limited hardware.

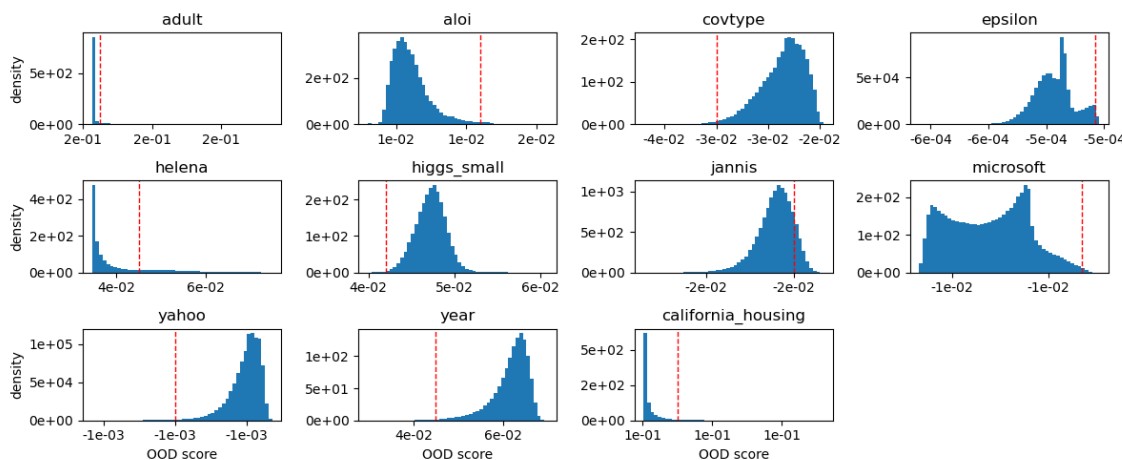

Figure 2: Histogram representing the OOD scores across various datasets. The red line is the threshold line that indicates whether the data is Out of Distribution or not.

## 7 RESULT AND EVALUATION

### 7.1 OOD DETECTION

Table 1: The OOD detection settings. Performances are results of model trained with linear regression ($r^2$) and logistic regression (accuracy). When OOD dataset is separated and used as test dataset in ($^b$) the performance of the model is decreased. OOD case in the epsilon ($^*$) dataset cannot be identified.

| Dataset | Detectors | Norms | OOD threshold | Accuracy without OOD [a] | | Accuracy with OOD [b] | |
| --- | --- | --- | --- | --- | --- | --- | --- |
| | | | | Train | Test | Train non-OOD | Test OOD |
| Adult | OpenMax | l2 | 0.162800 | 0.782561 | 0.783089 | 0.797838 | 0.267408 |
| Helena | OpenMax | l1 | 0.045000 | 0.194532 | 0.196626 | 0.146428 | 0.047553 |
| Jannis | TemperatureScaling | l1 | -0.020000 | 0.561953 | 0.563982 | 0.577274 | 0.474213 |
| Higgs small | OpenMax | l1 | 0.042000 | 0.622216 | 0.616879 | 0.622667 | 0.568493 |
| Aloi | OpenMax | l1 | 0.016000 | 0.260648 | 0.232546 | 0.328289 | 0.071186 |
| Epsilon $^*$ | TemperatureScaling | l2 | -0.000523 | 0.898635 | 0.897230 | 0.898700 | 0.893338 |
| Covtype | TemperatureScaling | l1 | -0.035000 | 0.604674 | 0.604373 | 0.603514 | 0.435439 |
| California Housing | OpenMax | l1 | 0.110000 | 0.333448 | 0.180680 | 0.477136 | -6.033595 |
| Year | OpenMax | l2 | 0.045000 | 0.168659 | 0.167098 | 0.169745 | -0.714197 |
| Yahoo | TemperatureScaling | l1 | -0.001460 | 0.325685 | 0.326275 | 0.325577 | -0.297845 |
| Microsoft | TemperatureScaling | l1 | -0.008300 | 0.045648 | 0.044109 | 0.047334 | -0.841856 |

Table 1 shows significant differences between the two settings. Without OOD (Table 1 , Section a), the training and test results are comparable. However, when used as test data, the OOD reduces the performance of the models. OOD leads to a 20% decrease (Table 1 , Section b) in performance between training and test results for the classification task, and a negative r2 for the regression task. The OOD separation process is

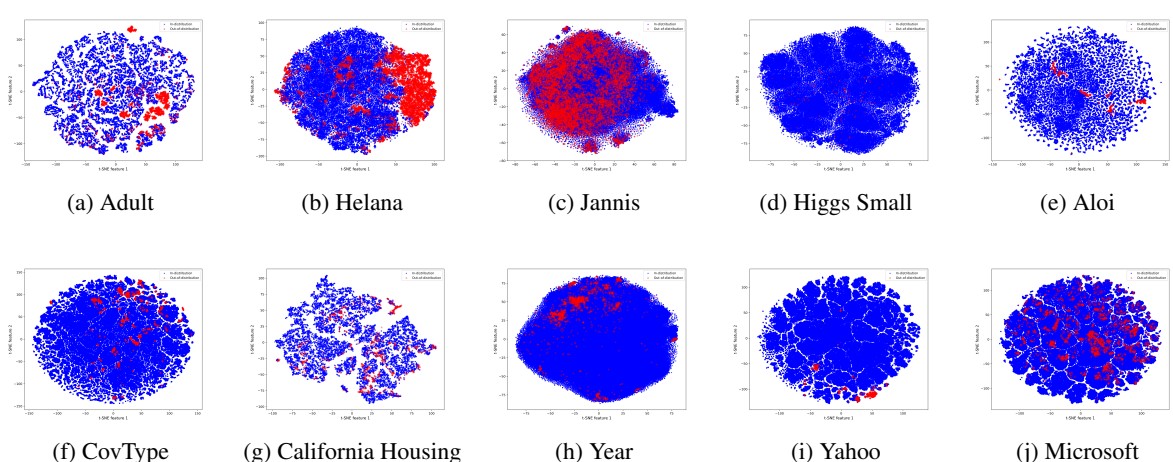

| (a) Adult | (b) Helana | (c) Jannis | (d) Higgs Small | (e) Aloi |
| (f) CovType | (g) California Housing | (h) Year | (i) Yahoo | (j) Microsoft |

Figure 3: OOD Visualisation. (.) or blue color is the in distribution data (ID), (+) or red is out of distrobution data (OOD). In all dataset, except for Jannis, it is clearl that OOD fills empty space between ID. Jannis data is evenly distributed, the visualisation capture neither ID nor OOD.

Table 2: Experiment result. F1 score for classification and RMSE for regression. Datasets with (*) mean a regression problem. Models with ($^c$) are contrastive learning based models.

| | AD↑ | HE↑ | JA↑ | HI↑ | AL↑ | CO↑ | CA*↓ | YE*↓ | YA*↓ | MI*↓ |
|---|---|---|---|---|---|---|---|---|---|---|
| FT-T | 0.782 | 0.153 | 0.572 | 0.738 | 0.407 | - | 0.867 | **6.461** | - | - |
| DCN2 | 0.744 | 0.129 | 0.542 | 0.710 | 0.414 | 0.58 | 2.602 | 7.054 | 0.645 | 0.746 |
| GrowNet | 0.465 | - | - | 0.685 | - | - | 0.969 | 7.605 | 1.01 | 0.769 |
| ResNet | 0.652 | 0.10 | 0.574 | 0.753 | 0.437 | 0.694 | 0.892 | 6.496 | **0.639** | **0.736** |
| MLP | 0.508 | 0.146 | 0.561 | 0.753 | 0.326 | 0.617 | 0.894 | 6.488 | 0.657 | 0.741 |
| AutoInt | 0.78 | 0.133 | 0.549 | 0.719 | 0.401 | 0.608 | 0.89 | 6.673 | - | 0.739 |
| TabR-MLP | 0.688 | 0.165 | 0.541 | 0.753 | 0.429 | 0.688 | 2.677 | 2e5 | 1.285 | 0.79 |
| TCL$^c$ | **0.831** | **0.154** | **0.575** | **0.758** | **0.447** | **0.880** | **0.843** | 6.491 | 0.652 | 0.738 |
| Scarf$^c$ | 0.720 | 0.00 | 0.122 | 0.308 | 0.00 | 0.091 | - | - | - | - |
| SubTab$^c$ | 0.714 | 0.146 | 0.504 | 0.602 | 0.322 | 0.59 | 1.012 | 6.668 | 0.656 | 0.744 |

visualized in Figure 2, which presents histogram graphs of the transformed data with detectors. The Epsilon dataset shows two peaks, indicating complexity, and outliers cannot be detected. In contrast, the Microsoft dataset shows a performance decrease with OOD. Figures 3 show the results of separating in-distribution (ID) and out-of-distribution (OOD) data using TSNE in an unsupervised manner. The figures demonstrate that OOD data fills the empty space between ID data, indicating a strong presence of OOD. Table 1, Figure 2, and Figure 3 show strong indications of the existence of out-of-distribution (OOD) data.

## 7.2 MODELS PERFORMANCE

Table 2 shows the results of the experiment. Overall, TCL outperforms other models, while the performance of the other models is comparable across various datasets. There are some exceptions where specific models underperform relative to others. For instance, GrowNet performs below average on the adult dataset, DCN V2 underperforms on the California Housing dataset, and GrowNet also underdelivers on the Yahoo dataset.

Table 3: Experiment result of TCL compared to GBDT. F1 score for classification and RMSE for regression. Datasets with (*) mean a regression problem. Model with ($^c$) means contrastive learning based model, models with ($^x$) mean GBDT based models.

| | AD↑ | HE↑ | JA↑ | HI↑ | AL↑ | CO↑ | CA*↓ | YE*↓ | YA*↓ | MI*↓ |
|---|---|---|---|---|---|---|---|---|---|---|
| TCL$^c$ | 0.831 | **0.154** | **0.575** | **0.758** | **0.447** | **0.880** | 0.843 | **6.491** | **0.652** | 0.738 |
| Lightgbm$^x$ | 0.591 | 0.080 | 0.432 | 0.609 | 0.177 | 0.219 | 0.848 | 6.565 | 0.661 | 0.740 |
| CatBoost$^x$ | **0.927** | 0.152 | 0.533 | 0.718 | - | 0.753 | **0.827** | 6.622 | 0.655 | **0.733** |
| XGB$^x$ | 0.925 | 0.127 | 0.532 | 0.739 | 0.328 | 0.700 | 0.845 | 6.867 | 0.654 | 0.739 |

Table 4: Table of training duration in second of each dataset. Datasets with (*) means a regression problem. All model except for TCL are trained with GPU. TCL were trained with CPU

| | AD↓ | HE↓ | JA↓ | HI↓ | AL↓ | CO↓ | CA*↓ | YE*↓ | YA*↓ | MI*↓ |
|---|---|---|---|---|---|---|---|---|---|---|
| FT-T | 1027 | 130 | 155 | 94 | 1205 | - | 88 | 1290 | - | - |
| ResNet | 2.1e+02 | 32 | 21 | 56 | 44 | 618 | 15 | 236 | 284 | 950 |
| TCL | 15 | 23 | 23 | 38 | 40 | 330 | 7 | 240 | 620 | 820 |

In contrast, TCL stands out by outperforming other models on most datasets, particularly in classification problems. Nevertheless, TCL's performance in regression problems is not significantly behind that of the top models..

When compared with the GBDT method, TCL outperforms in most datasets, especially in the classification problem, see Table 3. Compared to other classification problems, the adult dataset has a relatively higher score across other datasets. This shows that adult dataset does not require generalisation during prediction, which also explains why TCL performs under CatBoost. CatBoost dominant in 4 datasets beat any other GBDT algorithm.

## 7.3 TRAINING DURATION

Table 4 displays the training duration for the best three deep learning models. Each model has unique characteristics and training steps, and all seven models (FT-T, DCN 2, GrowNet, ResNet, MLP, AutoInt, TabR-MLP) underwent extensive tuning. The Yahoo and Microsoft datasets required 5 days to complete the entire parameter-tuning process. For FTT and restnet, a single training time was sampled once the tuning process was completed. TCL, which involve unsupervised training, The time recorded is time for each model to stabilize their loss with a 256 batch size, which is around 15 epochs. it is clear that TCL has a short training time.

## 7.4 EFFICIENCY EVALUATIONS

Table 5: A speed/accuracy trade off matrix $T = \frac{P}{t}$ where $P$ performance matrix used and $t$ is time in second required. A higher result is better. Datasets with (*) mean a regression problem.

| | AD | HE | JA | HI | AL | CO | CA* | YE* | YA* | MI* |
|---|---|---|---|---|---|---|---|---|---|---|
| FT-T | 0.00076 | 0.0012 | 0.0037 | 0.0079 | 0.00034 | - | 0.013 | 0.00012 | - | - |
| ResNet | 0.0031 | 0.0031 | 0.027 | 0.013 | 0.0099 | 0.0011 | 0.075 | 0.00065 | 0.0055 | 0.0014 |
| TCL | 0.055 | 0.0066 | 0.025 | 0.028 | 0.0199 | 0.0026 | 0.16 | 0.00064 | 0.0024 | 0.0016 |

Table 5 shows that TCLs are dominant. FT-Transformer and ResNet produce a good F1 and RMSE score; however, they take more time to train. In **FT-Transformer**, multiple attention heads process numeric and

categorical features separately before combining them. The model includes four types of layers that grow exponentially, leading to resource-intensive computations. **ResNet** employs parallel calculations across multiple convolutional layers (SubNet), using three identical SubNets, one of which is highly filtered. In contrast, TCL has a simpler architecture akin to MLP, achieving a high speed/accuracy trade-off. **TCL** features narrow layers for both the encoder and decoder, each with one hidden layer and one normalization layer, resulting in fewer layers than ResNet. However, TCL's pair operation for loss calculation doubles its training time.

### 7.5 TCL Efficiency Evaluation

Our TCL has undergone significant algorithm modifications, making the original similarity loss function inapplicable. We compare TCL with SubTab, which employs a similarity function for tabular contrastive learning, to evaluate TCL's efficiency.

Table 6 shows that the dot product applied on TCL consistently faster compared to similarity distance function applied to similar contrastive learning under SubTab. The efficiency gain from using the dot product supports our decision to incorporate it into the TCL model. Implementation of the entire original data matrix for representation removes matrix splitting as used in common contrastive learning. Implementation of dot product removes the requirement to calculate more complex similarity scores. Both algorithms were evaluated under CPU.

Table 6: Table of training duration in second of each dataset. Datasets with (*) mean a regression problem. TCL uses the dot product, and SubTab uses the similarity function. Both algorithms were evaluated under CPU.

|  | AD↓ | HE↓ | JA↓ | HI↓ | AL↓ | CO↓ | CA*↓ | YE*↓ | YA*↓ | MI*↓ |
|---|---|---|---|---|---|---|---|---|---|---|
| TCL | 15 | 23 | 23 | 38 | 40 | 330 | 7 | 240 | 620 | 820 |
| SubTab | 1400 | 1700 | 2400 | 2800 | 3400 | 1.8e+06 | 640 | 1.5e+04 | 3.6e+04 | 4e+04 |

## 8 Conclusion

The choice of models for tabular datasets with out-of-distribution (OOD) data depends on the user's needs and available resources. TCL outperforms other heavier models for classification problems on OOD while maintaining efficiency. RestNet and FT-Transformer perform well on many datasets, but these models require more resources, which may not always be feasible. It is worth noting that TCL was trained on a CPU, and RestNet and FT-T were trained on a GPU. This makes TCL available for more users than other models that require more training resources. Both RestNet and TCL can be options for fine-tuning and serving as head-to-head comparison models.

Although TCL has shown promising results, there are opportunities for potential enhancement. A continual learning can be proposed to improve performance. Further optimization of the contrastive learning process can be studied to achieve even greater efficiency. Additionally, there is a need to explore TCL's performance on a wider range of domain-specific tabular datasets. Furthermore, it is crucial to investigate TCL's interpretability, as this is important for many real-world applications.

## 9 Reproducibility

The code for this work can be found online (anonymized, 2024) (submitted as a supplementary file). The dataset is also available online and can be downloaded using the information provided in the citation.

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

# A APPENDIX

---

**Algorithm 1** OOD Detection and Train-Test Separation

---

**Require:** Dataset $D$, OOD detection methods $M = \{OpenMax, TemperatureScaling\}$
**Ensure:** In-distribution dataset $D_{in}$, Out-of-distribution dataset $D_{ood}$
  **for** each method $m \in M$ **do**
    Apply $m$ to $D$
    Obtain transformed data $D_m$
  **end for**
  **for** each $D_m$ **do**
    Visualize histogram of $D_m$
    Manually set threshold $t_m$ based on histogram
  **end for**
  Initialize $D_{in} \leftarrow \emptyset$, $D_{ood} \leftarrow \emptyset$
  **for** each sample $x \in D$ **do**
    **if** $\forall m \in M : D_m(x) < t_m$ **then**
      $D_{in} \leftarrow D_{in} \cup \{x\}$
    **else**
      $D_{ood} \leftarrow D_{ood} \cup \{x\}$
    **end if**
  **end for**
  Validate OOD separation using linear regression
  Compare model performance: $f(D_{in})$ vs $f(D_{ood})$
  **return** $D_{in}, D_{ood}$

---

**Algorithm 2** Tabular Contrastive Learning (TCL) Algorithm

---

**Require:** Tabular dataset $\mathcal{D}$, encoder $E$, decode $P$, batch size $N$, temperature $\tau$
  **for** each epoch **do**
    **for** each batch $\mathcal{B} \in \mathcal{D}$ **do**
      **for** each sample $x_i \in \mathcal{B}$ **do**
        Create augmented views $x_i^1, x_i^2 = \text{Augment}(x_i)$
      **end for**
      $\mathcal{B}_{\text{aug}} = \{x_i^1, x_i^2 | i = 1, \ldots, N\}$
      **for** each $\tilde{x}_i \in \mathcal{B}_{\text{aug}}$ **do**
        $x^e = E(\tilde{x}_i)$                                          ▷ Encode
        $x^p = P(x^e)$                                         ▷ Decode
      **end for**
      **if** inference **then return** $x^e$
      **end if**
      **for** $i = 1$ to $2N$ **do**
        $i^+ = (i + N) \bmod 2N$                             ▷ Pair index
        $L_t = L_r + L_c + L_d$
      **end for**
      $L_{tN} = \frac{1}{N} \sum_1^N L_t$
      Update $E$ and $P$ by optimizing $L_{tN}$
    **end for**
  **end for**

---