# OpenReview forum: "DEALING WITH OUT OF DISTRIBUTION IN PREDICTION PROBLEM"
_ICLR.cc/2025/Conference — ICLR 2025 Conference Withdrawn Submission_

### Official Review · Reviewer_cczR · 2024-11-01

**Soundness:** 1
**Presentation:** 1
**Contribution:** 1
**Rating:** 1
**Confidence:** 4

**Summary:**

This paper tackles the OOD issue for tabular data by using a framework consisting of contrastive learning. To separate in-domain and out-of-domain samples for experiments, the authors adopt OpenMax and Temperature Scaling with manually assigning the thresholds. The proposed TCL shows higher speed-accuracy trade-off scores, which was only trained on a CPU.

**Strengths:**

1. The proposed method is easy to understand, with a clear problem statement and definition.
2. Simaltaneously considering the OOD problem with efficiency may be an interesting direction, especially in the realm of large-scale models.

**Weaknesses:**

Generally, this paper requires extensive improvements in terms of motivations, related works, proposed method, as well as the experiments. Detailed comments are provided below:

1. The paper presentation should be improved. In the abstract, for instance, the issues in existing OOD works do not appear novel. Extensive fine-tuning and experimental trials are not unique to this specific OOD problem but relate to many general issues. The authors need to outline issues more specific to the OOD problem.
2. The review of related works requires significant improvement. There is a substantial body of existing work on OOD in tabular domains; however, the authors only discuss two methods in Sec. 2.1 and one other in L38, with the most recent paper referenced from 2017. The authors should put more effort into the literature review, for example, [1].
3. The proposed TCL employs a common strategy: contrastive learning with two views generated from a sample for self-supervised learning, followed by a head for classification/regression. This framework has been used in various contrastive learning-based tabular models, such as SubTab, which modifies contrastive targets with slicing techniques for tabular structures. The performance is not superior to the selected baselines, which limits both the novelty and effectiveness of the proposed approach.
4. The selected baselines are based on the work of Gorishniy et al., 2021, which was published in 2021. Since then, many advanced tabular prediction models have emerged and should be included in this paper, such as those summarized in [2]. Note that some of these models serve as tabular foundation models and may not experience the OOD issues described in this paper.
5. The authors used CPU(s) to train their model but used an H100 GPU for the baselines, which is confusing. I didn't see any specific design element that would make the proposed method more efficient; on the contrary, matrix augmentation appears to require more computational resources. Although Table 4 presents training durations, it is unclear how this table was created. For example, is the parameter space the same across all models?

[1] Multi-Class Data Description for Out-of-distribution Detection. KDD 2020.

[2] A Survey on Self-Supervised Learning for Non-Sequential Tabular Data. ACML 2024.

**Questions:**

Please see above comments.

---

> ### Author Response · Authors · 2024-11-13
>
> ```
> The paper presentation should be improved. In the abstract, for instance, the issues in existing OOD works do not appear novel. Extensive fine-tuning and experimental trials are not unique to this specific OOD problem but relate to many general issues. The authors need to outline issues more specific to the OOD problem.
> ```
>
> We will revise this later. Thank you for your input.
>
> ```
> The review of related works requires significant improvement. There is a substantial body of existing work on OOD in tabular domains; however, the authors only discuss two methods in Sec. 2.1 and one other in L38, with the most recent paper referenced from 2017. The authors should put more effort into the literature review, for example, [1].
> ```
>
> Our related work is actually more than this; however, we've done a substantial page deduction due to the page limit. Thank you; we will add this to the revision.
>
> ```
> The proposed TCL employs a common strategy: contrastive learning with two views generated from a sample for self-supervised learning, followed by a head for classification/regression. This framework has been used in various contrastive learning-based tabular models, such as SubTab, which modifies contrastive targets with slicing techniques for tabular structures. The performance is not superior to the selected baselines, which limits both the novelty and effectiveness of the proposed approach.
> ```
>
> We are using full matrix representation. We have done an upcoming experiment with better results, and we will release it this week. We also include XGB, Catboost and Lightgbm.
>
>
> ```
> The selected baselines are based on the work of Gorishniy et al., 2021, which was published in 2021. Since then, many advanced tabular prediction models have emerged and should be included in this paper, such as those summarized in [2]. Note that some of these models serve as tabular foundation models and may not experience the OOD issues described in this paper.
> ```
>
> We started this project in January, while the survey paper was listed in February. We are not aware of this. Our goal is to develop an algorithm that is light enough to be used by a general user without a GPU but capable of nearing or exceeding the capability of a user with a GPU. The methods mentioned in the survey paper are based on MLP and transformer. While we do not cover the methods mentioned in the survey paper, we include base MLP and Transformer.
>
>
> ```
> The authors used CPU(s) to train their model but used an H100 GPU for the baselines, which is confusing. I didn't see any specific design element that would make the proposed method more efficient; on the contrary, matrix augmentation appears to require more computational resources. Although Table 4 presents training durations, it is unclear how this table was created. For example, is the parameter space the same across all models?
> ```
>
> Our distinctive feature is the use of a full matrix representation in the design of TCL. This approach eliminates the need for data slicing and matrix multiplication. To optimize for CPU compatibility, we refine the loss function by employing only dot matrices and narrow layers. The core concept of contrastive learning involves adding noise and enhancing generalization. By applying this strategy, we can achieve performance comparable to larger models that typically demand more powerful hardware. We will add parameters specification on the next revision.

---

> > ### Comment · Reviewer_cczR · 2024-11-23
> > **Follow-up comments**
> >
> > Thank the authors for their response. Given that the updated manuscript still has significant issues and the responses do not address my concerns, I would retain my score accordingly.
> >
> > > Revision for the abstract.
> >
> > Thanks for the revision. However, the modified version still looks vague to me. Specifically, if the authors aim to tackle the issue that existing works require GPU to train, this is still not specific to the OOD problem, i.e., any deep learning tasks have this issue.
> >
> > > Revision for the related work.
> >
> > The current version looks like a brief introduction for each method, which needs to be linked to provide more paper presentation quality. The authors still do not discuss the relations between each method and the issues (only saying a critical area is very vague).
> >
> > > Novelty and effectiveness of the proposed approach.
> >
> > I do not get the novel idea of "using full matrix representation" as replied by the authors. The authors replied that the upcoming results yield better performance; still, there is no detail of this (e.g., what did the authors modify?).
> >
> > > The start date of the project.
> >
> > According to the ICLR policy, only the papers published after this July could not be considered. Therefore, the authors acknowledge their outdated baselines.

---

> > > ### Author Response · Authors · 2024-11-23
> > >
> > > Dear cczR23
> > >
> > > Thank you for taking the time to give feedback.
> > > ```
> > > Thanks for the revision. However, the modified version still looks vague to me. Specifically, if the authors aim to tackle the issue that existing works require GPU to train, this is still not specific to the OOD problem, i.e., any deep learning tasks have this issue.
> > >
> > > ```
> > > I understand the issue, and you are correct. However, our primary focus here is OOD (Out-of-Distribution), which can be addressed through model generalization. Our area of expertise lies in tabular contrastive learning for general users. In this context, contrastive learning proves highly effective as it calculates loss from noisy representations. Traditional contrastive learning methods are often resource-intensive because they reinterpret tabular data as images, treating them accordingly. To address this, we developed TCL, a lightweight contrastive learning model. TCL efficiently handles OOD scenarios by adding 50% noise into the data.  TCL can handle non-OOD data with smaller noise levels (we are not covering this). We are actively working to demonstrate that contrastive learning is a powerful approach to tackling OOD challenges.
> > >
> > > ```
> > > The current version looks like a brief introduction for each method, which needs to be linked to provide more paper presentation quality. The authors still do not discuss the relations between each method and the issues (only saying a critical area is very vague).
> > > ```
> > > We cannot include additional explanations as we need to cover a wide range of related works. Instead, we provide concise descriptions for each method and organize them under subheadings, which we believe is sufficient for the reader. The document has already reached the 10-page limit. We do not elaborate on their weaknesses since most of the related works mentioned are incorporated into our own work. Additionally, we offer more detailed technical explanations on page 10 to highlight some of the best models.
> > >
> > > >*Table 5 shows that TCLs are dominant. FT-Transformer and ResNet produce a good F1 and RMSE score;
> > > however, they take more time to train. In FT-Transformer, multiple attention heads process numeric and
> > > categorical features separately before combining them. The model includes four types of layers that grow
> > > exponentially, leading to resource-intensive computations. ResNet employs parallel calculations across mul-
> > > tiple convolutional layers (SubNet), using three identical SubNets, one of which is highly filtered. In con-
> > > trast, TCL has a simpler architecture akin to MLP, achieving a high speed/accuracy trade-off. TCL features
> > > narrow layers for both the encoder and decoder, each with one hidden layer and one normalization layer, re-
> > > sulting in fewer layers than ResNet. However, TCL’s pair operation for loss calculation doubles its training
> > > time.*
> > > ```
> > > I do not get the novel idea of "using full matrix representation" as replied by the authors. The authors replied that the upcoming results yield better performance; still, there is no detail of this (e.g., what did the authors modify?)
> > > ```
> > > We add a pair of Linear and ReLU layers, which increases the training time of our TCL by 50% compared to the previous version.
> > > We will update it in our github code.
> > > ```
> > > According to the ICLR policy, only the papers published after this July could not be considered. Therefore, the authors acknowledge their outdated baselines.
> > > ```
> > > The methods discussed in the referenced paper are not included in this work. However, we have incorporated a new baseline featuring SubTab, SCARF, and TabR, which are established and widely cited approaches. Additionally, we have included the Gradient Boosted Decision Trees (GBDT) method, recognized as one of the most effective techniques for tabular data [1].
> > >
> > > We acknowledge that our update may not fully address all concerns; however, we have made every effort to respond comprehensively and meet the expectations of all reviewers.
> > > Regards.
> > >
> > >
> > > [1]  Grinsztajn, Léo, Edouard Oyallon, and Gaël Varoquaux. "Why do tree-based models still outperform deep learning on typical tabular data?." Advances in neural information processing systems 35 (2022): 507-520.

---

> > > > ### Comment · Reviewer_cczR · 2024-11-25
> > > >
> > > > Thank you for your follow-up response. If the authors modified their proposed method, it indicates that the manuscript is not ready to be accepted. Given this and the issues mentioned previously, I am inclined to the original score for this paper.

---

> ### Author Response · Authors · 2024-11-25
>
> Dear cczR.
>
> Thank you for your response.
> Initially, our primary objective was to develop an efficient method, even if it required a slight compromise on accuracy. However, feedback from reviewers indicated a lack of acknowledgement of this approach. Consequently, we shifted our focus towards prioritizing accuracy, even at the expense of efficiency. This adjustment explains the modifications made to the algorithm.
>
>
> ```
>  If the authors modified their proposed method, it indicates that the manuscript is not ready to be accepted.
> ```
> We believe, based on the ICLR 2025 rule mentioned on https://iclr.cc/Conferences/2025/AuthorGuide :
> >*You can upload revisions until the discussion period ends*
>
> Revision is allowed.
> We have made every effort to respond comprehensively and meet the expectations of all reviewers.
>
> Regards.

---

> > ### Comment · Reviewer_cczR · 2024-11-25
> >
> > Thank the authors for their response. The ICLR policy accepts the chance, but one of the main issues is the motivation as well as the readability. For instance, if the authors change the proposed method, where could we find the corresponding updates?
> >
> > In addition, if the updated method was just added some MLPs, this can be viewed as parameter study, where the authors did not do it. Specifically, if adding one MLP + ReLU works, what are the effects of adding more layers? However, the response does not resolve one of my important yet foundamental concerns about the novel design, the outdated experiments, and the paper presentation.

---

> > > ### Author Response · Authors · 2024-11-25
> > >
> > > Dear cczR,
> > >
> > > We sincerely value and appreciate your feedback and responses.
> > >
> > > We are not incorporating an MLP into the algorithm; instead, we are using a single Linear layer followed by a ReLU activation from PyTorch.
> > > ```
> > > nn.Linear()
> > > F.Relu()
> > > ```
> > > We may have overlooked mentioning this earlier. We doubled the size of the hidden layer, which resulted in increased training time. To enhance efficiency, we initially trimmed these layers to optimize the model. However, in response to the reviewers' concerns, we decide to expand our deep learning network.
> > > We are not introducing new concepts. The design remains unchanged, and the changes are solely focused on programming design.
> > >
> > > Our goal is to develop a deep-learning algorithm for tabular data with out-of-distribution (OOD) capabilities that is lightweight and accessible to general users without requiring specialized hardware. We chose contrastive learning (CL) due to its ability to introduce noise into the data. However, existing CL methods are computationally intensive. To address this, we designed TCL.
> > >
> > > Our novelty lies in the use of a full matrix representation and a dot matrix loss function, which together enable efficient performance while maintaining high-quality results compared to heavier models.
> > >
> > > We understand your concerns regarding issues like readability. However, we have thoroughly reviewed the ICLR 2025 documentation and are confident that our revisions comply with all the rules.
> > >
> > > Regards

---

### Official Review · Reviewer_a2pd · 2024-11-02

**Soundness:** 1
**Presentation:** 2
**Contribution:** 2
**Rating:** 3
**Confidence:** 3

**Summary:**

The article presents a method called tabular contrast learning (TCL) aimed at improving model performance on tabular datasets, particularly in the context of Out-of-Distribution (OOD) tabular data. The authors highlight TCL's efficiency in reducing computational costs and its potential for deployment in resource-limited environments. The paper discusses the model's training process and experimental results across 10 datasets.

**Strengths:**

1. The research addresses an important issue in machine learning—handling OOD data, which is crucial for ensuring model reliability and robustness.
2. TCL demonstrates a very efficient approach that reduces computational costs, making it viable for deployment in environments with limited resources.

**Weaknesses:**

1. The experimental results are not particularly impressive. While TCL shows high efficiency, it only achieved optimal performance on three datasets compared to FT-T and ResNet.
2. The paper lacks a comprehensive comparison of methods. It would benefit from including comparisons with GBDT methods (e.g., CatBoost[1], XGBoost[2]), other state-of-the-art deep learning models (e.g., TabR[3], ExcelFormer[4], Trompt[5]), and alternative self-supervised learning algorithms (e.g., Scarf[6], VIME[7]) besides SubTab.
3. More details should be included in the paper, such as the specific structure of TCL, the downstream classifier used, hyperparameters of the comparison methods, and whether experiments were conducted multiple times to present average results.
4. The experiment comparing dot product and Euclidean distance indicates that dot product is more efficient on the hardware they used. However, this work does not discuss how using these two distance metrics impacts the performance of TCL and how significant the computational time for distance calculation is within the entire TCL process.

[1] Liudmila Ostroumova Prokhorenkova, Gleb Gusev, Aleksandr Vorobev, Anna Veronika Dorogush, and Andrey Gulin. Catboost: unbiased boosting with categorical features. In NeurIPS

[2] Tianqi Chen and Carlos Guestrin. Xgboost: A scalable tree boosting system. In KDD

[3] Yury Gorishniy, Ivan Rubachev, Nikolay Kartashev, Daniil Shlenskii, Akim Kotelnikov, and Artem Babenko. Tabr: Tabular deep learning meets nearest neighbors in 2023. In ICLR

[4] Jintai Chen, Jiahuan Yan, Danny Ziyi Chen, Jian Wu: ExcelFormer: A Neural Network Surpassing GBDTs on Tabular Data. In KDD

[5] Kuan-Yu Chen, Ping-Han Chiang, Hsin-Rung Chou, Ting-Wei Chen, Tien-Hao Chang: Trompt: Towards a Better Deep Neural Network for Tabular Data. In ICML

[6] Dara Bahri, Heinrich Jiang, Yi Tay, Donald Metzler:Scarf: Self-Supervised Contrastive Learning using Random Feature Corruption. In ICLR

[7] Jinsung Yoon, Yao Zhang, James Jordon, Mihaela van der Schaar: VIME: Extending the Success of Self- and Semi-supervised Learning to Tabular Domain. In NeurIPS

**Questions:**

1. See weaknesses
2. What criteria did the authors use to select different detectors for OOD detection across various datasets?
3. Authors may want to assess the performance of TCL using the recently published benchmark, TabRed [1], which is a Benchmark of Tabular Machine Learning in-the-Wild with Real-World Industry-Grade Tabular Datasets.

[1] Ivan Rubachev, Nikolay Kartashev, Yury Gorishniy, Artem Babenko: TabReD: A Benchmark of Tabular Machine Learning in-the-Wild.

---

> ### Author Response · Authors · 2024-11-13
>
> ```
> The experimental results are not particularly impressive. While TCL shows high efficiency, it only achieved optimal performance on three datasets compared to FT-T and ResNet.
> ```
> We have done follow-up experiments with better results. We will upload it in the next revision. Thank you
>
> ```
> The paper lacks a comprehensive comparison of methods. It would benefit from including comparisons with GBDT methods (e.g., CatBoost[1], XGBoost[2]), other state-of-the-art deep learning models (e.g., TabR[3], ExcelFormer[4], Trompt[5]), and alternative self-supervised learning algorithms (e.g., Scarf[6], VIME[7]) besides SubTab.
> ```
>
> We have added GBDT to our experiment. We will release it in our revision We can not include every available method due to our limitation. We will add Scarf in our next revision. We will consider others, but probably for the next paper. thank you
>
>
> ```
> More details should be included in the paper, such as the specific structure of TCL, the downstream classifier used, hyperparameters of the comparison methods, and whether experiments were conducted multiple times to present average results.
> ```
>
> This work focuses on how to detect and predict out-of-distribution. TCL is part of the paper but not all about TCL. Our work uses a regular dataset, predicts OOD and proof, and predicts classification or regression problems on it. This is what real-world users actually seek. In addition, we add TCL as a prediction tool for general users without GPU.
>
> ```
> The experiment comparing dot product and Euclidean distance indicates that dot product is more efficient on the hardware they used. However, this work does not discuss how using these two distance metrics impacts the performance of TCL and how significant the computational time for distance calculation is within the entire
> ```
>
> Our distinctive feature is the use of a full matrix representation in the design of TCL. This approach eliminates the need for data slicing and matrix multiplication. To optimize for CPU compatibility, we refine the loss function by employing only dot matrices and narrow layers. The core concept of contrastive learning involves adding noise and enhancing generalization. By applying this strategy, we can achieve performance comparable to larger models that typically demand more powerful hardware.
>
>
> ```
> TCL process.What criteria did the authors use to select different detectors for OOD detection across various datasets?
> ```
> We tried many detection algorithms, but we only used algorithms that can empirically be proven by the next step mentioned in our paper after detection.
>
> ```
> Authors may want to assess the performance of TCL using the recently published benchmark, TabRed [1], which is a Benchmark of Tabular Machine Learning in-the-Wild with Real-World Industry-Grade Tabular Datasets.
> ```
> We will definitely consider this in our revision, but for now, we will add Scraf as the method gain more popularity.

---

### Official Review · Reviewer_QoLi · 2024-11-02

**Soundness:** 3
**Presentation:** 2
**Contribution:** 2
**Rating:** 5
**Confidence:** 3

**Summary:**

This study focuses on detecting and predicting OOD data in tabular datasets. It provides methods for identifying OOD data, guidance on evaluating OOD selections, and introduces Tabular Contrast Learning (TCL), a technique optimized for tabular predictions.
TCL seems to be more computationally efficient than baseline models, making it suitable for general users with limited computational power. The study also compares TCL with existing approaches, emphasizing both accuracy and efficiency.

**Strengths:**

+ Multiple Dataset and Models

+ Impressing Experiments and Results

The manuscript tackles an important challenge in machine learning by focusing on out-of-distribution (OOD) data handling for tabular datasets, and introduces Tabular Contrast Learning (TCL) as a novel solution. This is a well-defined problem with significant practical implications, especially for general users with computational limitations. The authors have provided a clear motivation for their work, particularly in making OOD detection and handling accessible without high-end hardware requirements. The proposed TCL method appears promising for addressing efficiency and accuracy in tabular OOD tasks.

**Weaknesses:**

However, it would benefit from a more detailed comparison with recent advancements in tabular contrastive learning. Notably, two works could provide relevant baselines and enhance the context for TCL.
- Best of Both Worlds: Multimodal Contrastive Learning With Tabular and Imaging Data, CVPR23
- TabContrast: A Local-Global Level Method for Tabular Contrastive Learning, NIPS23

**Questions:**

- Why is the Contrastive Federated Learning (CFL) reference in an anonymized link?
- Why does the title miss some keywords like Tabular Contrastive Learning?
- Why is there no experiment for baseline models running on CPU (Apple) to fairly compare with your TCL?

---

> ### Author Response · Authors · 2024-11-13
>
> ```
> Why is the Contrastive Federated Learning (CFL) reference in an anonymized link?
> ```
>
> -> It is due to the anonymized rule, correspondence to one or more author of this paper.
>
>
> ```
> Why does the title miss some keywords like Tabular Contrastive Learning?
> ```
>
> -> This work focuses on how to detect and predict out-of-distribution. TCL is part of the paper but not all about TCL. Our work uses a regular dataset, predicts OOD and proof, and predicts classification or regression problems on it. This is what actually real-world users seek.
>
>
> ```
> Why is there no experiment for baseline models running on CPU (Apple) to fairly compare with your TCL?
> ```
>
> -> According to the original paper, baseline models are best performed within GPU. To support that, we follow the original paper's architecture to fit their best setting. We have a follow-up experiment with a CPU with better results. Our point is to show that with only a consumer CPU, we can achieve similar results with advanced models trained with GPU.

---

> ### Comment · Reviewer_QoLi · 2024-11-25
>
> Thank you for your dedication and continuous effort in refining this work. Considering the competitive nature of this field, I strongly encourage the authors to explore additional related works (as the reviewers point out) and emphasize the unique innovations and advantages of your methods. Specifically, it would be beneficial to provide more details about TCL, highlight the distinctions between TCL and CFL or other comparable frameworks, and clearly articulate the motivation and compelling aspects that set the methods apart.

---

> > ### Author Response · Authors · 2024-11-25
> >
> > Dear QoLi,
> >
> > Thank you for your feedback and suggestions.
> >
> > As outlined in our latest update, we have included Scarf, TabR, and GBDT models as comparisons with the most recent established methodologies. This was done in response to the reviewers' feedback. Additionally, we have incorporated MCDD into our related work to align with the latest OOD studies.
> >
> > Unfortunately, we are unable to change the title as per your suggestion.
> >
> > This section of the abstract has been revised according to your suggestions.
> > >*Furthermore, the study introduces Tabular Contrast Learning (TCL), a technique specifically designed for tabular prediction tasks. While achieving better results compared to heavier models, TCL is more efficient even when trained without specialised hardware, making it useful for general machine-learning users with computational limitations. This study includes a comprehensive comparison with existing approaches within their best hardware setting (GPU) compared with TCL on common hardware (CPU), focusing on both accuracy and efficiency. The results show that TCL exceeds other models, including gradient boosting decision trees, contrastive learning, and other deep learning models, on the classification task.*
> >
> > Our study provides a comprehensive discussion on OOD, covering everything from A to Z. TCL represents one aspect of this work. Our goal is to help everyone understand how to manage OOD data, starting from raw data processing to the development of a predictive model. Due to space constraints (10 pages), we could only include a limited amount of information in the paper. Additional algorithms are provided in the appendix, and we maintain an active GitHub page for further resources. We hope this will be sufficient.
> >
> > In our related work, we provide reasons that distinguish between TCL and CFL.
> > >*CFL focuses on collaborative learning across silos. In this study, we study learning from local data with OOD, a problem that is yet to be explored by CFL CFL focuses on a Federated learning network, while ours is common tabular data.  CFL, while exhibiting a similar name, uses partial data augmentation as part of the federated learning concept. TCL  did not come from a similar base understanding of CFL that mention data in a silo similar to partial data augmentation in original contrastive learning for the image. *
> >
> > We acknowledge that our update may not fully address all concerns; however, we have made every effort to respond comprehensively and meet the expectations of all reviewers.
> >
> > Regards.

---

### Official Review · Reviewer_UNbE · 2024-11-04

**Soundness:** 2
**Presentation:** 3
**Contribution:** 1
**Rating:** 1
**Confidence:** 5

**Summary:**

The paper applies an existing method (Contrastive Federated Learning, or CFL) to tabular data prediction tasks. The authors use 10 toy tabular datasets to evaluate this method and compare with a set of deep learning-based baselines on various classification and regression metrics.

**Strengths:**

# Overall assessment

In general, the writing in the paper could be improved considerably. The paper is missing references to many relevant works in the tabular prediction literature, and fails to compare to many relevant baselines (most notably, GBDT methods like XGBoost, LightGBM, and CatBoost). The datasets selected for the empirical comparisons are not appropriate for this study, as these datasets do not display distribution shift, while the authors do not use or acknowledge several existing benchmarks for OOD prediction in tabular data.

# Major comments

* Many of the main claims in the abstract lack support or are difficult to verify -- for example, "Addressing OOD data requires extensive fine-tuning and experimental trials" and "Deep learning has been sug-
gested as a solution and has shown significant improvements".

* The paper is missing references to many relevant works, including:
  - Malinin, Andrey, et al. "Shifts 2.0: Extending the dataset of real distributional shifts." arXiv preprint arXiv:2206.15407 (2022).
  - Gardner, Josh, Zoran Popovic, and Ludwig Schmidt. "Benchmarking distribution shift in tabular data with tableshift." Advances in Neural Information Processing Systems 36 (2024).
  - Liu, Jiashuo, et al. "On the need for a language describing distribution shifts: Illustrations on tabular datasets." Advances in Neural Information Processing Systems 36 (2024).

  All three of the above papers propose benchmarks for OOD-related tabular tasks. These would be much more appropriate than the 10 toy datasets selected for the empirical studies.

* The paper is also missing references to several relevant empirical studies for tabular data, including:
  - Gardner, Josh, Zoran Popovic, and Ludwig Schmidt. "Subgroup robustness grows on trees: An empirical baseline investigation." Advances in Neural Information Processing Systems 35 (2022): 9939-9954.
  - Grinsztajn, Léo, Edouard Oyallon, and Gaël Varoquaux. "Why do tree-based models still outperform deep learning on typical tabular data?." Advances in neural information processing systems 35 (2022): 507-520.
  - Kadra, Arlind, et al. "Well-tuned simple nets excel on tabular datasets." Advances in neural information processing systems 34 (2021): 23928-23941.

  In particular, the first two studies above suggest that GBDTs are perhaps the most appropriate baseline for this study, but they are omitted from the study completely.

* The paper frames its use of contrastive learning as a novel contribution, however, there are two problems with this:
  1. Contrastive learning has already been widely used for tabular data; See e.g. (Bahri, D., Jiang, H., Tay, Y., & Metzler, D. (2021). Scarf: Self-supervised contrastive learning using random feature corruption. arXiv preprint arXiv:2106.15147.) and the survey of (Rabbani, Shourav B., Ivan V. Medri, and Manar D. Samad. "Attention versus Contrastive Learning of Tabular Data--A Data-centric Benchmarking." arXiv preprint arXiv:2401.04266 (2024).) In particular I would note that its performance has generally not been competitive with strong baselines and adoption of the method has been limited as a result.
  2. The paper simply applies an existing method (Contrastive Federated Learning, or CFL) to tabular data. This is not sufficient for acceptance, given the concerns with the empirical evaluations described above.

# Minor comments

* More detail should be provided on the datasets in the main text (note that I suggest completely changing the datasets to more appropriate distribution shift datasets). In particular, the authors should explain why these generic tabular datasets (e.g. Adult, California Housing) are appropriate for studying the very specific task of OOD detection. Some descriptive statustucs about the datasets would also be useful.

# Typos etc.

* The paper is quite difficult to read in places due to grammatical issues. I would suggest having a native English speaker proofread the manuscript, or using a grammar checking tool.
* Definition 4: "eucledian"

**Weaknesses:**

See above.

**Questions:**

See above.

---

> ### Author Response · Authors · 2024-11-13
>
> ```
> are not appropriate for this study, as these datasets do not display distribution shifts,
> ```
> We deliberately use a common database. The reason behind this is In the real world, users can only use the databases that are available. For that reason, we also cover detection and steps to be taken for OOD.
> ```
> Many of the main claims in the abstract lack support or are difficult to verify -- for example, "Addressing OOD data requires extensive fine-tuning and experimental trials" and "Deep learning has been sug- gested as a solution and has shown significant improvements".
> ```
> We will add this in our revision. Thank you.
>
> ```
>  The paper is missing references to many relevant works, including:
> ```
> We will add this in our revision. Thank you.
>
> ```
> The paper is also missing references to several relevant empirical studies for tabular data, including:
> ```
> Our work focuses on how to detect and predict out-of-distribution. TCL is part of the paper but not all about TCL. Our work uses a regular dataset, predicts OOD and proof, and predicts classification or regression problems on it. Adding this will max out page limitation or deduct technical details of our paper.
>
> ```
> The paper frames its use of contrastive learning as a novel contribution, however, there are two problems with this:
> ...
> The paper simply applies an existing method (Contrastive Federated Learning, or CFL) to tabular data. This is not sufficient for acceptance, given the concerns with the empirical evaluations described above.
> ```
> CFL focuses on a Federated learning network, while ours is common tabular data. In CFL, while exhibiting similar name,  they use partial data augmentation as part of federated learning concept. In our CFL we use a complete full raw matrix representation. Our TCL also did not came from base understanding of CFL that mention about data in a silo similar to partial data augmentation in original contratstive learning for image. TCL is part of the paper, the whole paper is A-Z discussing about OOD treatment for general user.

---

> > ### Comment · Reviewer_UNbE · 2024-11-25
> >
> > Acknowledging that I have received and reviewed the author response. I will retain my score.

---

> > > ### Author Response · Authors · 2024-11-25
> > >
> > > Dear UNbE,
> > > Thank you for your comments and previous feedback.
> > >
> > > Based on your previous feedback, we have made extensive revisions to our paper. We have incorporated additional citations to substantiate our claims, included SCARF for a comprehensive contrastive learning comparison, and expanded our discussion to cover tree-based models. Finally, we distinguish between TCL and CFL in our related work.
> > >
> > > Regards.

---

### Author Response · Authors · 2024-11-22

Dear UNBe, QoLi, a2pd, cczR.

We revised the paper according to feedback from earlier reviews. We improved the transition from the introduction to enhance the overall flow.
```
...When a model encounters OOD data, its performance can significantly decrease. Improving the model’s performance in dealing with OOD can be achieved through generalisation by adding noise, which can be easily done with deep learning. However, many advanced machine learning models are resource-intensive and designed to work best with specialized hardware (GPU), which may not always be available for common users with hardware limitations. To provide a deep understanding and solution on OOD for general users, this study explores detection, evaluation, and prediction tasks within the context of OOD on tabular datasets using common consumer hardware (CPU). It demonstrates how users can identify OOD data from available datasets and provide guidance on evaluating the OOD selection through simple experiments and visualizations. ...
```
In related work, we mention our differences with CFL.
```
CFL focuses on collaborative learning across silos. In this study, we study learning from local data with OOD, a problem that is yet to be explored by CFL CFL focuses on a Federated learning network, while ours is common tabular data.  CFL, while exhibiting a similar name, uses partial data augmentation as part of the federated learning concept. TCL  did not come from a similar base understanding of CFL that mention data in a silo similar to partial data augmentation in original contrastive learning for the image.
```
We have incorporated MCDD [1] in our related work and explained why it was not utilized. Our experiments feature gradient-boosting decision trees, SCARF [2], and TabR [3], as suggested by reviewers. Below are the updated results from our experiments:

|          | AD    | HE    | JA    | HI    | AL    | CO    | CA*   | YE*      | YA*   | MI*   |
|----------|-------|-------|-------|-------|-------|-------|-------|----------|-------|-------|
| FT-T     | 0.782 | 0.153 | 0.572 | 0.738 | 0.407 | -     | 0.867 | 6.461    | -     | -     |
| DCN2     | 0.744 | 0.129 | 0.542 | 0.71  | 0.414 | 0.58  | 2.602 | 7.054    | 0.645 | 0.746 |
| GrowNet  | 0.465 | -     | -     | 0.685 | -     | -     | 0.969 | 7.605    | 1.01  | 0.769 |
| ResNet   | 0.652 | 0.1   | 0.574 | 0.753 | 0.437 | 0.694 | 0.892 | 6.496    | 0.639 | 0.736 |
| MLP      | 0.508 | 0.146 | 0.561 | 0.753 | 0.326 | 0.617 | 0.894 | 6.488    | 0.657 | 0.741 |
| AutoInt  | 0.78  | 0.133 | 0.549 | 0.719 | 0.401 | 0.608 | 0.89  | 6.673    | -     | 0.739 |
| TabR-MLP | 0.688 | 0.165 | 0.541 | 0.753 | 0.429 | 0.688 | 2.677 | 2.00E+05 | 1.285 | 0.79  |
| TCL      | 0.831 | 0.154 | 0.575 | 0.758 | 0.447 | 0.88  | 0.843 | 6.491    | 0.652 | 0.738 |
| Scarf    | 0.72  | 0     | 0.122 | 0.308 | 0     | 0.091 | -     | -        | -     | -     |
| SubTab   | 0.714 | 0.146 | 0.504 | 0.602 | 0.322 | 0.59  | 1.012 | 6.668    | 0.656 | 0.744 |

Our recent experiments demonstrate that TCL surpasses other methods in solving classification problems. We also conducted a comparison between TCL and GBDT, following the recommendations by UNbE and the research by Grinsztajn et al. [4].

|          | AD    | HE    | JA    | HI    | AL    | CO    | CA*   | YE*   | YA*   | MI*   |
|----------|-------|-------|-------|-------|-------|-------|-------|-------|-------|-------|
| TCL      | 0.831 | 0.154 | 0.575 | 0.758 | 0.447 | 0.88  | 0.843 | 6.491 | 0.652 | 0.738 |
| Lightgbm | 0.591 | 0.08  | 0.432 | 0.609 | 0.177 | 0.219 | 0.848 | 6.565 | 0.661 | 0.74  |
| CatBoost | 0.927 | 0.152 | 0.533 | 0.718 | -     | 0.753 | 0.827 | 6.622 | 0.655 | 0.733 |
| XGB      | 0.925 | 0.127 | 0.532 | 0.739 | 0.328 | 0.7   | 0.845 | 6.867 | 0.654 | 0.739 |

TCL outperform GBDT based model in classification.
Unfortunately, we can not change the title as QoLi suggested.

Regard,

[1] Dongha Lee, Sehun Yu, and Hwanjo Yu. Multi-class data description for out-of-distribution detection.In Proceedings of the 26th ACM SIGKDD International Conference on Knowledge Discovery & Data Mining, KDD ’20, page 1362–1370, New York, NY, USA, 2020b. Association for Computing Machinery. ISBN 9781450379984. doi: 10.1145/3394486.3403189. URL https://doi.org/10.1145/
3394486.3403189.

[2] Dara Bahri, Heinrich Jiang, Yi Tay, and Donald Metzler. SCARF: SELF-SUPERVISED CONTRASTIVE LEARNING USING RANDOM FEATURE CORRUPTION. In ICLR 2022 - 10th International Conference on Learning Representations, 2022.

[3] Yury Gorishniy, Ivan Rubachev, Nikolay Kartashev, Daniil Shlenskii, Akim Kotelnikov, and Artem Babenko. Tabr: Tabular deep learning meets nearest neighbors in 2023, 2023. URL https://arxiv.org/abs/2307.14338.

[4] Grinsztajn, Léo, Edouard Oyallon, and Gaël Varoquaux. "Why do tree-based models still outperform deep learning on typical tabular data?." Advances in neural information processing systems 35 (2022): 507-520.

---

### Author Response · Authors · 2024-11-29

Dear UNBe, QoLi, a2pd, cczR.

As the discussion date approaches, We would like to express our gratitude for all the feedback received. We have revised (2nd) our original paper to address the comments from all reviewers. This includes improvements to the flow, the addition of significant experiments incorporating the latest methods, and an updated literature review.

We believe we have addressed all reviewers' comments. While one reviewer noted that the revisions are major, we have ensured that these changes comply with the guidelines for ICLR 2025. Please consider that our paper serves as a comprehensive guide for out-of-distribution (OOD), not just TCL.

Finally, we kindly request the reviewers to reassess their scores based on this revision.


Regards

---

### Note · Authors · 2025-01-29

**Comment:**

Dear ICLR,
I believe the third version, which has been heavily edited from the 1st version, is not covered in the review comment; I understand the decision and would like to withdraw if possible. I want to submit it to another conference.

Regard,

**Withdrawal Confirmation:**

I have read and agree with the venue's withdrawal policy on behalf of myself and my co-authors.